# Psychophysical Risk Perceptions and Sleep Quality of Medical Assistance Team Members in Square Cabin Hospitals: A Repeated Cross-Sectional Study

**DOI:** 10.3390/healthcare10102048

**Published:** 2022-10-17

**Authors:** Qianlan Yin, Xiaoqin Shao, Rong Zhang, Jiemei Fan, Wei Dong, Guanghui Deng

**Affiliations:** 1Department of Psychology, Navy Medical University, Shanghai 200433, China; 2Department of Nursing, No. 989 Hospital of Joint Logistics Support Forces of PLA, Luoyang 471031, China; 3Department of Critical Care Medicine, Jinling Hospital, Nanjing 210002, China; 4Department of Psychology, Fudan University, Shanghai 200011, China

**Keywords:** psychophysical risk, sleep quality, risk perceptions, emotional states, Medical Assistance Team Members, lockdown

## Abstract

**Objective:** This study aimed to evaluate the association between the perceptions of psychophysical risks and sleep quality of Medical Assistance Team Members (MATMs) in Square Cabin Hospitals. **Methods:** Repeated cross-sectional data collection was conducted in Square Cabin Hospitals during two large-scale lockdowns. The first wave was sampled from MATMs dispatched to Wuhan and the second was from MATMs dispatched to Shanghai. Participants completed online questionnaires comprised of the Risk Perception Questionnaire (RPQ), Positive and negative emotions scale (PANAS), and Sleep Quality Scale (SQS), measuring the psychophysical risk perceptions about the MATMs’ current work, emotional states, and sleep quality. Changes across two waves of data collection were statistically parsed using the exploratory factor analysis and regression models. **Results:** Data of 220 participants from first-wave samples [S1] and 300 from second-wave samples [S2] were analyzed. Participants reported more worries about physical risks, such as inadequate protection methods and being infected, and S1 rated higher on all risks compared with S2 (as the biggest *p*-value was 0.021). Across the different situations, the dominant emotional states of MATMs were positive; a higher level of psychophysical risk perceptions, negative emotional states, and poor sleep quality were consistently interrelated. The psychophysical risk perceptions predicted sleep quality. Negative emotions as a state variable intensified the relationship between physical risk perceptions and sleep quality (b_indirect effect_ = 1.084, bootstrapped CI = [0.705, 1.487]). **Conclusions:** The results provide important evidence that MATMs’ higher level of psychophysical risk perceptions associated with negative emotions could indicate worse sleep quality.

## 1. Introduction

The most efficient method of combating the pandemic illness is through lockdowns (which include stay-at-home orders, curfews, quarantines, cordon sanitaire, and similar societal restrictions) and mobile medical services [1,2]. Since the first lockdown in Wuhan, China’s lockdown protocol has been implemented in several large, populated cities. Recognizing the Square Cabin Hospitals’ role in providing COVID-19 patients with preventive treatment, as well as their speedy resolution of the issue of inadequate beds in existing hospitals, is one of the lessons that have to be learned. Square Cabin Hospitals may admit patients much more quickly, lowering the rate at which mild and moderate cases turn into severe and critical cases [3,4]. Medical resources and personnel were inadequate in the early phases of the epidemic [5]. The Chinese government therefore implemented a medical aid strategy in which neighboring cities sent out medical teams to relieve the strain on overburdened local healthcare systems. A total of 42,322 professional healthcare workers (HCWs) from throughout the nation joined the medical aid team, as stated, to help with the lack of healthcare workers [6].

However, the harsh working conditions have increased the likelihood that HCWs on a medical aid team (referred to as MATMs) can have traumatic symptoms, such as burnout, secondary traumatic stress, anxiety, and so forth [6]. As we all know, maintaining physical health depends on the physiological process of sleep, which is also highly vulnerable to stress. MATMs must adapt to significant changes in the surrounding environment. However, stress brought on by changes may result in symptoms, such as sleep deprivation and increased wakefulness, which raises the risk of sleep disorders including insomnia, daytime sleepiness, nightmares, and daytime dysfunction, among others [7]. Pappa et al. (2020) conducted a meta-analysis to look at the prevalence of sleep disorders among medical professionals that treat COVID-19 patients; the study found that 38.8% of the population had such disorders [8]. According to a prior study, full-time frontline hospital employees had a 3.14 greater likelihood of complaining about insomnia and substantially worse general health compared to normative data [9]. Meanwhile, symptoms of insomnia and general health were unrelated to age, job experience, educational level, and gender but individual psychological characteristics had an unneglectable effect on their well-being. A cross-sectional study investigated the sleep quality of 1036 MATMs dispatched to Wuhan and showed that 52.4% of participants reported symptoms of a sleep disorder [10]. The study came to the further conclusion that exposure status and duration of labor were the primary variables determining sleep state, which had indirect impacts on sleep status via the mediator of regulatory emotional self-efficacy. Additionally, a study of 543 frontline medical staff in Wuhan during the peak of COVID-19 showed a noteworthy increase in the prevalence of negative emotions and sentiments among the medical staff, along with poor overall sleep quality [11]. Therefore, it would be a reasonable hypothesis to state that the psychological responses of the MATMs to the harsh environment may have a direct impact on the quality of their sleep.

The difficulties that MATMs must overcome include the need to expose themselves to hazards, a fear of infection, the need to execute novel activities in environments with limited resources, and a lack of familiarity with team members. The psychological and behavioral responses of MATMs, including their ability to sleep and carry out everyday activities, may be complicated by incorrect perceptions of the hazards. As a result, how MATMs perceived unexpected and increased risks while fighting the pandemic may have a significant impact on how well they sleep and how well they fight the disease. Indeed, there has been a lot of scholarly interest in the medical and psychological risks that HCWs face. The main elements of HCWs’ perceived risks, according to a previous study, were personal health risks, risks to others’ health, social isolation, and acceptance of risks [12]. A thorough investigation came to the conclusion that HCWs’ perceptions of risk could be clustered into six dimensions: personal safety (threats to life or body usually caused by irrational patients), physical function (injuries to physical health from illnesses), occupational exposures (exposures to virus and injuries from operations), psychosocial concerns (stigmas and other social judgments), organization safety (threats to self-interest caused by hospital regulations and management), and timing pressure (limited time for own pleasures), which provides a dimensional model derived from risk perception for medical work [13,14,15]. On the background of the COVID-19 epidemic, a study of 2078 HCWs in Italy reported that HCWs highly rated the risk perception of being infected, and 63.43% of them reported that they suffered from sleep disturbances [16]. Another study also reported how the stress of becoming infected was a significant contributor to sleep disturbances among HCWs who were treating COVID-19 patients [9]. Moreover, a retrospective study of 120 MATMs showed that the presentation of fear of being infected with COVID-19 was the main factor contributing to the occurrence of depression symptoms during quarantine [17]. Worries about high-intensity work, non-nuclear families, low hobbies, and irregular diet led to worse mental health in MATMs in Wuhan [18]. In particular, a high level of perception about their health and safety risks can influence the retention of HCWs within the workforce and their willingness to care for infected patients [19,20]. From a psychological perspective, a public health crisis that causes numerous losses in a short period harms psychological well-being, which is closely related to the perception of a public health crisis [21]. In this way, the subjective perceptions of MATMs play an important role in their ability to maintain psychological well-being under unexpected and risky situations.

So far, the data reported in the scientific literature on the COVID-19 pandemic, for the most part, comprises information gathered at a particular moment in time during the pandemic, allowing for an understanding of the connection between psychophysical hazards and sleep quality. Although it was challenging to track the relationship’s dynamic changes considering the mobility of the MATMs, the repeated cross-sectional study could also provide insights at a group-level analysis. People who perceive varying levels of pandemic threats will be more sensitive to and temperamental about governmental-enforced measures, such as lockdowns, during COVID-19, while MATMs act in the opposite way, working in Square Cabin Hospitals, unlike most individuals who stay alone at home. As we expected, MATMs’ emotional moods and exposure status had a greater impact on their psychophysical risks, sleep quality, and relationships. Our study compared the survey findings provided by the two MATM groups, who were the first to be sent to the lockdown city and its Square Cabin Hospitals, with a view to examining the changes in the connection. Especially, both groups encountered unexpected and non-experienced challenges and stressors centered on the context of two similar significant and protracted lockdowns.

The present study investigated the MATMs in Square Cabin Hospitals; specifically, their special experiences during work, based on the two similar large-scale lockdowns in China. We focused on the perceptions of psychophysical risks related to their work and sleep quality. Furthermore, we considered their varied emotional states based on subjective evaluations. We hypothesize the following: (I) the psychophysical risk, sleep quality, and the emotional state of MATMs would be significantly different between the two-wave surveys due to the varied exposure status; (II) the psychophysical risk, sleep quality and emotional states could be interrelated across the waves; (III) a higher level of psychophysical risk perception could predict poorer sleep quality.

## 2. Materials and Methods

This study was approved by the Ethical Committee of Navy Medical University. The participants gave their consent online by ticking the agreement at the bottom of the informed consent page. After ticking, they entered the interface of questionnaires.

### 2.1. Participants

The cohort of this study was two samples of MATMs. The detailed procedures of recruiting are described later in this paper. For samples of Wave 1 (S1), a total of N = 288 MATMs received the invitation and opened the survey. However, only 220 questionnaires qualified concerning the subject’s job criteria (only involving nurses and doctors) and the standard time for completing the questionnaires (set at 500–1500 ms). For samples of Wave 1 (S2), a total of 400 MATMs opened the survey and 336 returned qualifying questionnaires. A total of 36 MATMs, who were assigned to Wuhan cabin hospitals, were excluded. After receiving confirmation from the supervisors of the teams, we ensured all of the MATMs had no recorded mental health problems or several sleep problems before being dispatched. The effective recruiting rates were 76.34% and 75.00% for each wave sampling. There were no missing data and items for all the qualified data since the survey was composed of mandatory questions. A detailed description of the demographic differences between the two samples is presented in Table 1. In particular, counting from the day that they arrived at the hospitals to the day that they were being investigated allowed us to roughly estimate the employment duration of our participants (when they still worked in the hospitals). We gave the participants the option of choosing between less than 1 month and more than 1 month, taking into account that our wave sampling started one month following the lockdown. There was no discernible variation in the work time of the two-wave samples, as shown in Table 1, even though the MATMs arrived in distinct batches throughout both waves.

### 2.2. Procedure

A repeated cross-sectional data collection was conducted and is shown in Figure 1. The first wave was conducted after the lockdown of Wuhan in China (accessed on 27 January 2020) when the MATMs were assigned to Wuhan. Specifically, the first wave began on 20 February 2020, in Wuhan Square Cabin Hospital. The second wave was conducted after the lockdown of Shanghai in China (accessed on 1 April 2022) when the MATMs were assigned to Shanghai. The beginning date of the survey was 15 May 2022, in Shanghai Square Cabin Hospital. In our cohort, MATMs were all assigned from hospitals from other provinces and the MATMs in the second wave had no prior experience of being dispatched to work in Wuhan (to maintain the same baseline of the two wave samples). Moreover, the cohort of MATMs in this study had worked in Square Cabin Hospitals for at least half a month but less than one and a half months. The repeated cross-sectional study was conducted online through the WeChat platform. Data were collected using the systematic sampling method. The link to the survey was sent by the directors of the dispatched teams in their WeChat groups. The WeChat groups in which the participants opened the survey were logged and the logs were confirmed after the data were returned to the surveyors. In this way, the survey was sent to the targeted participants. The first page of the survey informed them that this survey aimed at investigating their valuation of perceived risks related to work, current mood, and sleep quality, and ensured their voluntary participation and data safety. The two-wave survey continued within five days from when they were posted. There was no systematic error causing drop-out throughout the survey and no independent or overlapped samples.

### 2.3. Measures

#### 2.3.1. Risk Perception Questionnaire (RPQ)

The questionnaire on the MATMs’ risk perception for COVID-19 referred to the risk perception questionnaire of medical staff [15]. The questionnaire was self-rated and included 12 questions about different dimensions: personal safety risk (three items), physical function risk (three items), occupational exposure risk (one item), psychosocial evaluation risk (three items), organizational risk (one item), and time pressure (one item). The rating potentiality of risk was divided into five grades from “never” to “almost always” and was assigned 1–5 points in turn. The higher the score for each dimension, the higher the HCWs’ awareness of the work-relative risk. The total score represents the general level of risk perception.

Positive and negative emotions scale (PANAS).

The Chinese version of the emotional self-rating scale (PANAS- Positive and negative emotions scale) was adopted in the emotional self-rating scale, which was verified by Chinese scholars to have cross-cultural consistency [22]. The internal consistency of this scale had good reliability of 0.87 and was widely used. The scale contained 20 words describing emotions, including 10 positive words and 10 negative words. The participants were asked to evaluate the emotional intensity they experienced in their current state on a scale of 5, for which 1 meant “very slight or no”; 2 for “a little”; 3 for “moderate“; 4 for “relatively strong”, and 5 for “extremely strong”. Adapting to the needs of the survey, seven negative and seven positive emotions were extracted in the emotional assessment to form an effective version of the emotional scale.

#### 2.3.2. Sleep Quality Scale (SQS)

The Sleep Quality Scale is a self-administrated questionnaire that assesses sleep quality and is adapted from the Chinese vision of The Pittsburgh Sleep Quality Index (PSQI) [23]. A total of 4 questions from the PSQI were selected for assessing difficulty in falling asleep, sleep disturbances, daily energy, and total sleep quality. The PSQI is a self-rated questionnaire, which assesses sleep quality and disturbances over a 1-month time interval. Each question was weighted equally on a 1–4 scale, respectively presenting different choices; higher scores indicate worse sleep quality.

### 2.4. Statistical Analysis

First, the mean scores of indicative answers on specific items describing the valuations of MATMs working in different periods were analyzed. As a preliminary analysis, we verified the factor structure of the instruments using parallel and exploratory factor analysis of the data from S1 and confirmatory factor analysis of the data from S2. The model fit of the proposed model was assessed using a chi-square, comparative fit index (CFI > 0.80), root mean square error of approximation (RMSEA; <0.08) and standardized root mean square residual (SRMR; <0.08) [24]. We also verified the measurement invariance of the two samples, in which a change in CFI > 0.01 and in RMSEA > 0.015 was considered significant [25]. Structural equation models were estimated with a full information maximum likelihood approach and effects coding identification. The internal consistency of all scales was calculated as McDonald’s omega and Cronbach alpha coefficient. The correlation between the scales was calculated, and the difference between the two wave samplings using Fisher’s test was examined. Finally, we used the general linear regression model and a mediation model to predict sleep quality. All the analyses utilized Amos and SPSS with the extension of an OMEGA and PROCESS macro [26,27]. The criterion of the *p*-value is 0.05 and the bootstrapped CIs were calculated with 5000 samples.

## 3. Results

Examining the circumstance at the phenomenological level would be more suited to highlight the situational distinctions between two-wave sampling. Therefore, the analysis was begun with an examination of the disparities at the level of each valuation. The data presented in Table 2 includes the results of the items from RSQ, PANAS, and SQS. As the results show, all of the items of RPQ were rated higher by S1 than S2 (as the biggest *p*-value was 0.021). Using 2.5 as the medial level of risk perception, S1 reported more worries about work-related risks, especially the inadequate protection methods and the low chance of being cured after infection. Meanwhile, the nervous (t = 2.975, *p* = 0.003) and afraid (t = 4.259, *p* < 0.001) feelings of S1 were higher than S2, while the enthusiastic (t = −2.341, *p* = 0.020), interested (t = −3.947, *p* < 0.001), and proud (t = −3.702, *p* < 0.001) feelings of S1 were significantly lower than S2. Moreover, the sleep qualities of S1 tended to be worse than S2 in terms of falling asleep (t = 4.995, *p* < 0.001), easily waking up at night (t = 5.708, *p* = 0.001), and less daily energy (t = 2.362, *p* < 0.001).

For RPQ, the independent *t*-test showed that differences existed between the two samples in all items. To decompose the dimension of the scales, a factorial analysis indicated two major factors that accounted for 60.91% of the total variances. One factor was composed of the first seven items enclosing the physical risks related to work. The other factor was composed of the last five items enclosing the psychosocial risks related to work. Additionally, the parallel analysis indicated a better fit of the two-factor model (ꭓ^2^(53) = 407.787, *p* < 0.001, CFI = 0.889, RMSEA = 0.112, 90%CI [0.102, 0.123], SRMR = 0.173) than the one-factor model (ꭓ^2^(54) = 486.643, *p* < 0.001, CFI = 0.864, RMSEA = 0.124, 90%CI [0.114, 0.134], SRMR = 0.055). Therefore, we kept the two-factor structure for the RPQ with physical risks and psychosocial risks in the next analysis. Meanwhile, we configured their models according to the scale dimension for PANAS (with two dimensions) and SQS (with one dimension). To compare the measurement invariance models between Wave 1 and Wave 2, we used a parallel analysis for each scale structure. The results are presented in Table 3.

Parallel analysis of RPQ, PANAS, and SQS indicated that these scales showed a structure, loading, and intercept invariance on two-wave sampling. As the criterion of the significant model was a change in CFI > 0.01 and in RMSEA > 0.015, there were no significant changes in the models of RPQ and PANAS, while SQS did not fit the model of equal intercepts indicating the changes in structure between the two waves. Given the fact that the scales of PQR and SQS demonstrated an intercept invariance in two-wave measures, further analysis of their relationships could be justified at the mean levels of variables. However, there should be caution for the SQS. As the results illustrate in Table 4, the McDonald’s Omega and Cronbach alpha of the scales above 0.70 indicate a higher consistency across items and waves [28]. Hence, we calculated the total scores of the subscales formed based on the testified factor-model structures and explored their inner consistencies and outer relationships. Notably, we utilized the partial correlation analysis to control the effects of differences in the gender and occupations of the MATMs. After this control, the correlation coefficients of risk perceptions and negative emotions, psychosocial risk perceptions and sleep quality, and negative emotions and sleep quality, were significant across the different waves (*p*-values were less than 0.01). Furthermore, the strength of these associations changed across the waves. Risk perceptions and positive emotions turned out to be significantly and negatively related in Wave 2, and the strength of the association between psychosocial risk perception and negative emotions was significantly lessened (according to the |z| > 1.96). However, the associations between sleep quality and other variables remained the same strength. Therefore, we speculated that positive emotions were more dominant than negative emotions in MATMs in Wave 2 samples. Together, these results suggested that the sleep quality of MATMs would be worse as the negativity of psychophysical valuations increased.

Based on the meaningful relationship between risk perceptions, negative emotions, and sleep quality, we further explored the hypothesis that psychophysical valuations predicted the sleep quality of MATMs consistently over the two measurement waves. We evaluated two regression models using two-wave samples with age, gender, and occupation as controlled variables, and different psychosocial valuations as predictors according to the hypothesis. The results in Table 5 show that the psychophysical valuations had significant predictive effects on sleep quality; however, the waves affected the dominant predictors independently from the group differences in age, gender, and occupation. In Wave 1, worry about psychosocial risk was a significant and major contributor to worse sleep quality (b = 2.152, *p* = 0.047), while in Wave 2, worry about physical risk (b = 0.859, *p* = 0.010) accompanied by a negative emotional state (b = 0.337, *p* < 0.001) turned out to be the significant and major contributor to worse sleep quality. Although age was different in the two-wave samples, it only contributed to the sleep quality in Wave 2 and not to the risk perceptions and negative emotions (the minimal *p*-value was 0.089 for the effect on psychosocial risks). Hence, age was an independent predictor of sleep quality. To establish whether a negative emotional state could mediate the relationship between physical risk perceptions and sleep quality, we tested a mediation model utilizing the data of Wave 2 with age, gender, and occupation as controlled variables. As expected, the analysis showed the direct effects of physical risk perceptions on sleep quality (b = 0.865, bootstrapped CI = [0.354, 1.375]) and on negative emotions (b = 2.740, bootstrapped CI = [2.118, 3.332]), and the indirect effects of physical risk perceptions on sleep quality (b = 0.925, bootstrapped CI = [0.574, 1.275]) were statistically significant as the bootstrapped CIs excluded zero [29]. Therefore, emotion valuation as a state variable intensifies the relationship between physical risk perceptions on sleep quality.

## 4. Discussion

The present study aimed to examine how psychophysical risk perceptions were related to changes in the sleep quality of MATMs working in Square Cabin Hospitals. Using two-wave sampling during two big-scale lockdowns in China, our study found that across the different situations of lockdown, MATMs with negative emotional states consistently reported a higher level of psychophysical risk perceptions and poor sleep quality. Importantly, the psychophysical risk perceptions predicted the sleep quality in both samplings. More precisely, worry for psychosocial risk was the major contributor to poor sleep quality in MATMs during the Wuhan lockdown, while worry for physical risk connected with a negative emotional state of MATMs turned out to be the significant and major contributor to worse sleep quality during the Shanghai lockdown.

Overall, these results should be interpreted against the context of the lockdown situation in China at the time in which the study was conducted. Although the two waves of data collection had the same time course after the lockdown’s implementation, they captured different situations in China. The first wave study was conducted during the Wuhan lockdown in the time following the COVID-19 outbreak when the experience and equipment for epidemic protection work were insufficient. These issues were handled at the time of the second wave study conducted during the Shanghai lockdown; however, the issues of overwhelmed patients and the highly contagious virus still existed. To some extent, it is reasonable to assume that the atmosphere during the second wave could not be as intensive as that during the first wave. These differences were also reflected in the evaluation of the MATMs’ psychophysical risks, as S1 of our study reported more worries about their work than S2 reported, especially concerning the physical risk of being infected and inadequate protection. Moreover, S1 rated higher for negative emotions than S2, significantly in the evaluation of feeling nervous and afraid. Noticeably, both the S1 and S2 MATMs rated higher for positive emotions in their emotional evaluations of the present situation. We interpreted the results as MATMs of both waves were voluntarily there to aid in front-line work and were spirited by the mission; also, the MATMs expressed their determination and dedication to their work. However, the MATMs were not immune, themselves, from the psychophysical risks, thus they were expressing negative emotions and having sleep problems. The typical sleep problem in both waves was falling asleep. Meanwhile, S1 was more serious and reported easier waking up and less daily energy than S2. Hence, given the contrasting results, we could attain that the results of the MATMs’ valuations of psychophysical risks, emotional states, and sleep quality, varied across the environments and tended to be worse if the situation became more intense and/or deteriorated.

The measurements of psychophysical risks, emotional states, and sleep quality remained stable over the course of the repeated survey, and their associations with the variables we were interested in remained consistent. In previous studies, risk perception and a decrease in negative emotions varied with the development of the pandemic [15,30,31]. The physical risk of being infected and the psychosocial risk of less family time were perceived at the highest level in each dimension. Tension (feeling nervous), fear (feeling afraid), and worry (feeling upset) dominated the dimension of the MATMs’ negative emotions; alertness, determination, and enthusiasm were the dominant positive emotions. The psychophysical risk perceptions were negatively related to positive emotions and positive to negative emotions. These results were similar to the results found in a study investigating public samples concerning the emotion and risk perception of vaccines [32]. Different from prior studies on HCWs, our cohort of MATMs, as the first-line warriors for public health, were voluntarily in the hardest-hit areas regardless of the threat of infection. According to their accounts, their experiences fighting COVID-19 instilled in them a sense of responsibility to alleviate the patient’s suffering and a desire to coordinate efforts to protect the entire country from the virus [6]. S2 working in Shanghai cabin hospital showed more saliant pleasant feelings as a result of these spirits and encouragements. Negative emotions, on the other hand, were not dissipated. In comparison, S1 and S2 rated their terror differently (being afraid). Given the positive relationship between negative emotion and risk perceptions, we hypothesized that S1 experienced persistent fear of infection due to the virus’s contagious nature, unknown transmission modes, close contact with patients, and infection occurring in their colleagues, whereas S2 had some knowledge of COVID-19 and adequate protections, appearing more psychologically prepared for the virus, but they were still unaware of the contagious and dangerous variants and threatened by physical risks. As a result, the MATMs’ perceptions of dangers and job preparations may benefit from maintaining a happy mood and controlling emotional responses.

HCWs frequently experience issues with poor sleep quality. Increased dangers at work during the epidemic would make this worse. Numerous cross-sectional pieces of research and reviews have revealed that frontline HCWs experience sleep loss, sleep dysfunction, and sleep disorders [33,34,35,36]. A cross-sectional study containing 219 volunteers working as MATMs in Wuhan and 729 HCWs working in Ningbo (a city in China) found that the former group had double the ratio of insomnia as the latter [37]. Nevertheless, due to the difficulty of following up, repeated or extensive investigations of MATMs are rare. As a result, the repeated cross-sectional study design was adopted in our unique study to concentrate on the problem. We concentrated on the sleep quality of MATMs who were working in Square Cabin Hospitals and dealing with unique work schedules and exposure. Sampling from various times and locations revealed environmental variations, which ecologically helped us to understand the predictors of MATMs’ sleep quality. As predicted, significant changes in the two-wave data on sleep quality were discovered. Additionally, contrasting results demonstrated that risk perceptions were a major predictor of sleep quality, which was in line with earlier studies [14,35]. However, individual levels of a particular risk perception fluctuated depending on the situation and emotional shifts, which had an impact on the predictions. Despite the prevalence of positive emotions in our cohort, we still came to the conclusion that a low level of poor sleep quality in MATMs could be attributed to worry about physical risks that interacted with negative emotions, as shown in the data from S2, while a high level of poor sleep quality could primarily be attributed to psychosocial risks, which included worries about both personal and interpersonal relationships. Psychosocial risks can affect a worker’s psychological and physical health through a stress-mediated pathway [38]. Communicable disease outbreaks exacerbate these risks [39,40]. In our study, the MATMs, especially in S1, showed more worries about their leisure time and work performance, which was hard to balance and brought a lot of anxiety and pressure. Work overload during the pandemic, including patient numbers, caring work, and additional and unintended shifts, reduces the autonomy of HCWs to decide how to use their time [41]. This stress disrupted the MATMs sleep and led to a vicious cycle, which explained why the psychosocial risk perceptions strongly predicted the high level of poor sleep quality. Being relatively prepared during the Shanghai lockdown, the MATMs of S2, were generally less worried than S1, apprehended the situation, and thus only those whose emotional self-regulation worsened their risk perceptions constantly worried about their physical risks. Therefore, poor sleep quality was detected by the constant worries for physical safety in those MATMs feeling afraid or nervous. It was highlighted that material and psychological resources should be provided across all stages of the pandemic to enable MATMs to process difficult experiences. During the lockdown, to prevent the deterioration of MATMs’ sleep quality, there should be targeted psychological interventions aimed at MATMs’ emotional regulation and risk perceptions.

The implications of our study may help to improve understanding of the role of MATMs’ risk perceptions during lockdowns when they were working as medical volunteers in Square Cabin Hospitals. To better comprehend the many concerns regarding the unique environment and the emergence of emotional disorders, research carried out at various lockup inceptions may be helpful. Whereas the intensity of the pandemic is unstable and the virus is spreading, we have centered our research on the altered link between psychophysical perceptions and subjective sleep quality [42]. This study offers a chance to observe changes at the group level by gathering data at two lockdowns (although with two distinct populations). The results show that there was a rise in concerns about hazards, such as diseases, bad behavior, and an imbalance between work and life during the first lockdown. The findings also show a link between elevated perceived risk and poor sleep quality, with emotional control playing a role in mediating the correlations. This is a significant result since it gives health authorities advice on how to provide MATMs and other HCWs with psychological support. Namely, in future similar situations, the HCWs should be well encouraged and protected to secure their psychophysical safety; meanwhile, educating MATMs on how to appropriately regulate their emotions and deal with stress is important and indispensable, which should begin when they are dispatched. As such, interventions to improve their sleep quality might also be delivered simultaneously to improve their physical and psychosocial perceptions of their work and enhance the defensive work for the pandemic.

When evaluating the findings of this study, it is important to note a few limitations. First, the study’s selection of two distinct groups is not indicative of MATMs in general with regard to socio-demographic factors. In particular, the samples included a higher percentage of women and nurses than the overall population of MATMs, though the proportion of each group varies samely. The social-demographic parameters were controlled and not examined because the current study concentrated on the impact of psychophysical perceptions on sleep quality. To further investigate the effects of these variables, future studies should utilize a sizable sample of MATMs. Second, although the measures used in our study were validated and retested the consistency across the two samples, the scale for sleep quality was slightly unequal concerning the items. The reason might be that the short version of five items was insufficient to form a stable structure. However, the consistency of the scale still indicated a single factor of sleep quality. Therefore, it is crucial that a more comprehensive structure of sleep quality assessments be created for use in future studies on the pandemic. Last but not least, the heterogeneities of individuals in similar groups had no significant effects on the group-level analysis, and the repeated cross-sectional design of this study prevents the drawing of any inferences about causality, which is a major restriction. Additional proof for forecasting sleep quality and ideas for therapies should be obtained from longitudinal studies. In order to fully understand HCWs’ psychophysical risk perceptions and sleep quality, it is vital to work with expanded research collaboration and, to the greatest extent feasible, implement validated measures for the pandemic across diverse contexts.

## 5. Conclusions

This paper presents the results of one of the few studies conducted over two-time points that have focused on psychophysical risk perceptions and sleep quality of MATMs in Square Cabin Hospitals during large-scale city lockdowns. The results showed that MATMs’ perceptions of psychophysical risk were connected to their quality of sleep. For example, lower sleep quality may be predicted by a higher level of psychophysical risk perceptions linked to unpleasant emotions. Health officials should focus on ensuring MATMs’ psychophysical safety and highlight emotional guidelines in a new working environment. As a result, MATMs’ proper physical and psychological perceptions of their work may also increase the quality of their sleep, which will further strengthen their ability to defend against pandemics.

## Figures and Tables

**Figure 1 healthcare-10-02048-f001:**
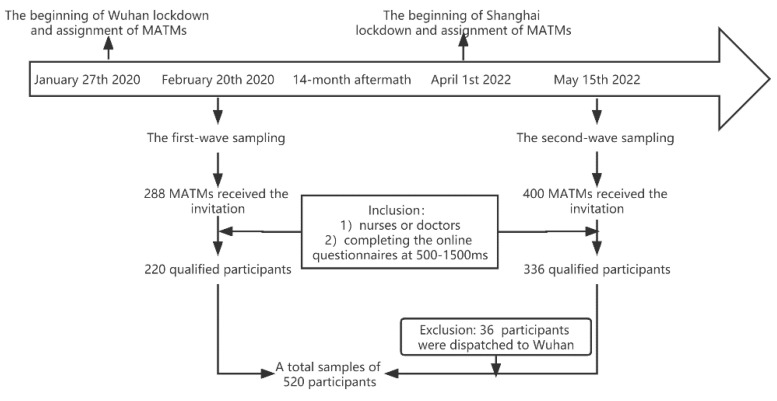
The flow chart of the data collection protocol.

**Table 1 healthcare-10-02048-t001:** Demographic characteristics of the samples of Wave 1 and Wave 2.

	Wave 1		Wave 2			
	Mean (sd)	N (%)	Mean (sd)	N (%)	t/χ^2^	*p*
Age/Gender					0.843	0.392
Men	35.66 (8.18)	38 (17.28)	34.44 (7.57)	43 (14.33)	0.695	0.489
Women	31.13 (6.50)	182 (82.73)	33.74 (6.27)	257 (85.67)	−4.231	<0.001
Total	31.91 (7.01)	220	33.84 (6.46)	300	−3.201	0.001
Nurses					17.713	<0.001
Yes		159 (72.27)		262 (87.33)		
No		61 (27.73)		38 (12.67)		
Work time					0.255	0.532
Less than 1 month		39 (17.73)		47 (15.67)		
More than 1 month		181 (82.27)		253 (84.33)		

**Table 2 healthcare-10-02048-t002:** Mean scores of each item from RSQ (a 5-point scale for none/often), PANAS (a 5-point for none/strong), and SQS (a 4-point scale for none/often).

Items	S1	S2	t	*p*
Risk Perception Questionnaire (RPQ)				
You often worry that there would be disputes in the medical process	2.81 ± 1.05	2.18 ± 0.82	7.428	<0.001
You often worry that people who are out of control would hurt you physically	2.89 ± 0.98	2.20 ± 0.80	8.579	<0.001
You often worry that high-level stress would affect your health	3.04 ± 1.05	2.53 ± 0.83	5.918	<0.001
You often worry that you might be infected due to contact with an infected person	3.21 ± 0.99	2.71 ± 0.76	6.284	<0.001
You often worry that, if infected, you would have a slim chance of being cured	2.62 ± 0.94	1.88 ± 0.78	9.589	<0.001
You often worry that you would be isolated from other people	2.59 ± 1.09	2.27 ± 0.85	3.577	<0.001
You often worry about being cut or stabbed by a contaminated sharp instrument during an operation	2.86 ± 0.99	2.27 ± 0.84	7.223	<0.001
You often worry about being unable to detect changes in the patient’s condition in time and delaying rescue	3.12 ± 1.04	2.64 ± 0.87	5.564	<0.001
You often worry about how your colleagues or patients perceive you negatively	2.48 ± 1.02	2.22 ± 0.87	3.068	0.002
You often worry about making mistakes at work	2.78 ± 0.97	2.59 ± 0.82	2.320	0.021
You often worry that there were inadequate medical resources, hospital protection, and isolation measures	2.91 ± 1.19	1.87 ± 0.77	11.427	<0.001
You often worry that you might have less time to spend with your family	3.20 ± 1.20	2.69 ± 0.91	5.237	<0.001
Positive and negative emotions scale (PANAS)				
Irritable	1.58 ± 0.84	1.58 ± 0.76	−0.085	0.933
Distressed	1.51 ± 0.83	1.42 ± 0.74	1.381	0.168
Upset	1.75 ± 0.91	1.82 ± 0.89	−0.933	0.351
Nervous	1.92 ± 0.9	1.69 ± 0.85	2.975	0.003
Guilty	1.31 ± 0.64	1.38 ± 0.71	−1.135	0.257
Afraid	1.61 ± 0.84	1.32 ± 0.68	4.259	<0.001
Jittery	1.50 ± 0.83	1.4 ± 0.77	1.407	0.160
Attentive	3.33 ± 1.29	3.51 ± 1.12	−1.652	0.099
Determined	3.56 ± 1.25	3.78 ± 1.02	−2.177	0.030
Alert	3.88 ± 1.14	3.92 ± 1.01	−0.442	0.659
Excited	2.23 ± 1.16	2.23 ± 1.13	−0.059	0.953
Enthusiastic	3.09 ± 1.35	3.35 ± 1.15	−2.341	0.020
Interested	2.65 ± 1.24	3.06 ± 1.09	−3.947	<0.001
Proud	3.26 ± 1.36	3.68 ± 1.16	−3.702	<0.001
Sleep Quality Scale (SQS)				
You have difficulty falling asleep	2.57 ± 1.11	1.85 ± 0.81	4.995	<0.001
You wake up easily at night	2.47 ± 1.07	1.61 ± 0.85	5.708	<0.001
You have less daily energy	1.87 ± 0.91	1.82 ± 0.68	0.380	0.705
You have poor sleep quality	2.27 ± 0.77	2.00 ± 0.65	2.362	0.022

**Table 3 healthcare-10-02048-t003:** Comparison of the measurement invariances model between Wave 1 and Wave 2.

Scale	Model	χ^2^	df	Δχ^2^	CFI	RMSEA	RMSEA 90%CI		ΔRMSEA
RPQ	**1**	**416.906**	**106.000**		**0.890**	**0.075**	**0.068**	**0.083**	
	2	454.591	117.000	37.685	0.880	0.075	0.067	0.082	0.000
	3	471.249	119.000	54.343	0.875	0.076	0.068	0.083	0.001
	4	613.253	131.000	196.347	0.829	0.084	0.078	0.091	0.009
PANAS	**1**	**646.046**	**154.000**		**0.875**	**0.079**	**0.072**	**0.085**	
	2	673.954	166.000	27.908	0.871	0.077	0.071	0.083	−0.002
	3	681.384	168.000	35.338	0.870	0.077	0.071	0.083	−0.002
	4	736.694	182.000	90.648	0.859	0.077	0.071	0.083	−0.002
SQS	**1**	**418.395**	**110.000**		**0.980**	**0.050**	**0.004**	**0.085**	
	2	452.483	115.000	34.088	0.906	0.090	0.065	0.117	0.040
	3	457.456	116.000	39.061	0.897	0.092	0.067	0.117	0.042
	4	475.377	121.000	56.982	0.865	0.091	0.069	0.113	0.041

Note: 1-Unrestricted; 2-Equal loadings; 3-Equal structure; 4-Equal intercepts. The bolded models were the reference model.

**Table 4 healthcare-10-02048-t004:** Correlation coefficients and reliability of RPQ, PANAS, and SQS across two periods.

	Index	Physical Risk	Psychosocial Risk	Negative Emotion	Positive Emotion	Sleep Quality	Omega	Alpha
Physical risk	Wave 1		0.821 **	0.508 **	0.127	0.279	0.891	0.891
	Wave 2		0.713 **	0.522 **	−0.146 *	0.397 **		
	z		−2.985	0.213	−3.076	1.495		
Psychosocial risk	Wave 1	0.821 **	1	0.607 **	−0.039	0.451 **	0.796	0.794
	Wave 2	0.713 **	1	0.465 **	−0.134 *	0.300 **		
	z	−2.985		−2.245	−1.073	−1.976		
Negative emotion	Wave 1	0.508 **	0.607 **	1	−0.067	0.412 *	0.901	0.898
	Wave 2	0.522 **	0.465 **	1	−0.114 *	0.517 **		
	z	0.213	−2.245		−0.531	1.503		
Positive emotion	Wave 1	0.127	−0.039	−0.067	1	0.034	0.874	0.870
	Wave 2	−0.146 *	−0.134 *	−0.114 *	1	−0.100		
	z	−3.076	−1.073	−0.531		−1.504		
Sleep Quality	Wave 1	0.279	0.451 **	0.412 *	0.034	1	0.744	0.713
	Wave 2	0.397 **	0.300 **	0.517 **	−0.100	1.000		
	z	1.495	−1.976	1.503	−1.504			

Note: * means *p*-value is below 0.05; ** means *p*-value is below 0.01; The partial correlation analysis controlled the effects of the differences in gender and occupation of the MATMs.

**Table 5 healthcare-10-02048-t005:** Results of the hierarchical multiple regression analysis predicting sleep quality.

Model	Predictors	Wave 1					Wave 2				
		b	sd	Beta	t	*p*	b	sd	Beta	t	*p*
1	Constant	11.303	3.808		2.968	0.005	6.312	1.393		4.533	0.000
	Age	−0.101	0.083	−0.249	−1.219	0.231	0.060	0.029	0.124	2.087	0.038
	Gender	0.195	1.354	0.036	0.144	0.886	0.232	0.707	0.026	0.328	0.743
	Occupation	−1.203	1.416	−0.236	−0.849	0.402	0.439	0.759	0.046	0.578	0.564
2	Constant	6.242	4.078		1.531	0.136	2.350	1.516		1.550	0.122
	Age	−0.110	0.076	−0.271	−1.441	0.160	0.054	0.025	0.111	2.186	0.030
	Gender	0.579	1.305	0.107	0.444	0.660	−0.027	0.608	−0.003	−0.045	0.964
	Occupation	−1.503	1.300	−0.295	−1.156	0.256	0.190	0.655	0.020	0.290	0.772
	Negative emotions	0.169	0.158	0.208	1.067	0.294	0.337	0.050	0.393	6.705	0.000
	Positive emotions	0.055	0.059	0.148	0.929	0.360	−0.018	0.027	−0.032	−0.642	0.521
	Physical risk	−1.024	0.833	−0.337	−1.229	0.228	0.859	0.333	0.193	2.581	0.010
	Psychosocial risk	2.152	1.039	0.606	2.071	0.047	−0.020	0.360	−0.004	−0.055	0.956

## Data Availability

The data presented in this study are available on request from the corresponding author. The data are not publicly available due to the involvement of personnel information.

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
