# Peer review of "Psychophysical Risk Perceptions and Sleep Quality of Medical Assistance Team Members in Square Cabin Hospitals: A Repeated Cross-Sectional Study"

_healthcare, 2022, doi:10.3390/healthcare10102048_

Round 1

Reviewer 1 Report

Dear Authors,

the study is interesting and bring some new insights on the psychophysiological consequences of MATMs working in cabin hospital during in Covid-19 pandemic However, the manuscript needs some improvements before considering it for further processing:

It is slightly confusing to divide the study into 2 waves as the samples were totally different. I would rather call it study 1 and study 2.  

Introduction

The introduction focuses too much on specific context. What I miss in the introduction is more reflection of psychophysiological consequences of stress and sleep disorders of medical staff in other studies.

Please add research hypotheses

There were quite some newcomers - how long they worked could significantly affect the results.

Another issue that could bias the results is the duration of their dispatchment to work in the cabin hospitals.

Measures

What is the internal consistency of RPO measure?

Results

I am not truly convinced that it is the most appropriate to compare the “waves” as the data comes from totally different samples.

The mean scores for all the items in RSO are relatively low. Any comment on it.?

The negative emotions were also scored relatively low. Any comment on it?

Refer in the discussion on other factors could potentially influence the sleep quality in such extreme situations as MATMs experienced e.g. what I miss in the measurement instrument is any question identifying how being dispatched from home could affect the emotional status of MATMs?

Author Response

Reviewer 1:

1)It is slightly confusing to divide the study into 2 waves as the samples were totally different. I would rather call it study 1 and study 2.  

Response:

Thank you for the insightful advice. We still prefer the original terminology above the ones that are frequently used in reports of repeated cross-sectional studies[1] after carefully examining the design of our study. Our cohort study's goal was to analyze population changes over time (also known as aggregate change over time), not individual changes, even though the samples were completely different at the individual level. Notably, our target audience consisted of MATMs who had been transferred from hospitals in other provinces and who had never before been sent to work in Square Cabin Hospitals. Instead of conducting two separate investigations, distinct people made up the study's two-wave samples, and relative analyses of population changes were also used.

[1] Wang, X., & Cheng, Z. (2020). Cross-Sectional Studies: Strengths, Weaknesses, and Recommendations. Chest, 158(1, Supplement), S65-S71. doi: https://doi.org/10.1016/j.chest.2020.03.012

2) The introduction focuses too much on specific context. What I miss in the introduction is more reflection of psychophysiological consequences of stress and sleep disorders of medical staff in other studies. Please add research hypotheses.There were quite some newcomers - how long they worked could significantly affect the results. Another issue that could bias the results is the duration of their dispatchment to work in the cabin hospitals.

Response:

We added some references that demonstrate the connections between mental stress and sleep disorders in HCW working at the front lines, such as Abdoli, N., Farnia, V., Jahangiri, S., Radmehr, F., Alikhani, M., Abdoli, P., and Brand, S. (2021). Hospital Staff Working During the COVID-19 Pandemic: Sources of Sleep Disturbances and Psychological Stress, Int J Environ Res Public Health, 18(12), doi:10.3390/ijerph18126289. The second paragraph of the introduction also added three more references. The introduction's final section included a list of three hypotheses.

We discussed how the length of dispatchment, in particular, allowed us to roughly estimate the duration of employment of our participants by counting from the day they arrived at hospitals to the day they were being investigated (when they still worked in the hospitals). Given that our wave sample began one month after the lockdown, we provided participants the option of choosing between less than one month and more than one month. Although MATMs arrived in different batches during each of the two waves, there was no appreciable difference in the work time of the two-wave samples, as indicated in Table 1.  

Measures

3) What is the internal consistency of RPQ measure?

Response:

  Table 4 shows the internal consistency of all questionnaires, including the RPQ, as determined using all samples. Additionally, we improved the Omega internal validity index for the repeated study in table 4.

4) I am not truly convinced that it is the most appropriate to compare the “waves” as the data comes from totally different samples.

Response:

  We have stressed this in the discussion and acknowledge the reviewer's worries about the repeative cross-sectional design's limitations. Actually, we would like to investigate the various effects of two-time dispatchments in the same samples, which was more convincing as a longitude study. However, we found these effects would be more lag-related and intricated, not to speak of the small number of qualified participants. Alternatively, a repeative cross-sectional design was more feasible for our purpose.

The study's goal was to describe the effects on MATMs caused by the first-hand knowledge of working in Square Capital Hospital. The target populations of the study were therefore the frontline workers who were initially sent to the area that was severely affected. In order to prevent the mixed effects of experiencing the stress conditions personally or not personally, no repeated sample was used in the final dataset. Instead, descriptive and inferential analyses for the relevant variables were conducted on a separate dataset, and results were cautiously combined at the population level. According to assessments, this method resulted in the correct and convincing conclusions.

5) The mean scores for all the items in RSQ are relatively low. Any comment on it.?The negative emotions were also scored relatively low. Any comment on it?

Response:

  Based on the background of our study conducted, there were many potential factors potentially contributed to the relatively lower scores for negative emotions and RSQ. As we explained in the discussion(-”Different from prior studies for HCWs, our cohort of MATMs as the first-line warriors for public health were voluntary to the hardest-hit areas regardless of the threat of infection. According to their accounts, their experiences fighting COVID-19 instilled in them a sense of responsibility to alleviate patients' suffering and a desire to coordinate efforts to protect the entire country from the virus(6). S2 working in Shanghai cabin hospital showed more saliant pleasant feelings as a result of the spirits and encouragements”), the heroic sensation of MATMs would be a key factor under such conditions and their volunteriness also represented their tolerance to the risks. Additionally, there would be plenty of helpful resources and favorable media coverage for our samples in China, which were demonstrated to be crucial for the psychological wellbeing of HCWs. However, the paper skipped over a lot of other important issues. We hope it will be debated in the future with additional research.  

6) Refer in the discussion on other factors could potentially influence the sleep quality in such extreme situations as MATMs experienced e.g. what I miss in the measurement instrument is any question identifying how being dispatched from home could affect the emotional status of MATMs?

Responses:

We appreciate you asking the question because it was intriguing to us. Further data analysis revealed that the time of dispatch was substantially associated to risk perceptions for spending less time with families (as a part of psychosocial risks) rather than sleep quality and negative emotions. However, it is important to keep in mind that because the dispatched time was limited to a narrow range of 1 month and did not vary significantly, there was not enough information to draw any conclusions in this study. Therefore, we did not separately discuss this question and still used focused on relations between the total scores of perceived risks and emotional status. Meanwhile, we appealed in the article that a future study should be designed to explore whether being dispatched from home could affect emotional status of MATMs.  

Reviewer 2 Report

Study of quality of life in health care workers during confinement in the first 2 waves of the pandemic. Appropriate sample size and correct methodological design. Although the conclusions are quite predictable, they are very well supported due to the 2 cutoffs in both study samples.

It is necessary for the authors to explicitly indicate whether the individuals in the samples previously suffered from sleep disorders (hypersomnias, parasomnias,...) or whether this issue was investigated or not in order to avoid possible biases since the prevalence of sleep disorders is high among healthcare personnel.

I recommend the authors to review this paper (which evaluates the suffering of patients with COVID during hospital isolation): 

López López, Carla; Vicho de la Fuente, Noelia; López Reboiro, Manuel Lorenzo; Blanco Hortas, Andrés; López Castro, José. The other face of the pandemic: psychosocial perspective of patients hospitalized by COVID-19. Presence. 2022; 18: e13972. Available at: http://ciberindex.com/c/p/e13972 [accessed: 08/09/2022].

Author Response

It is necessary for the authors to explicitly indicate whether the individuals in the samples previously suffered from sleep disorders (hypersomnias, parasomnias,...) or whether this issue was investigated or not in order to avoid possible biases since the prevalence of sleep disorders is high among healthcare personnel. I recommend the authors to review this paper (which evaluates the suffering of patients with COVID during hospital isolation):  

López López, Carla; Vicho de la Fuente, Noelia; López Reboiro, Manuel Lorenzo; Blanco Hortas, Andrés; López Castro, José. The other face of the pandemic: psychosocial perspective of patients hospitalized by COVID-19. Presence. 2022; 18: e13972. Available at: http://ciberindex.com/c/p/e13972 [accessed: 08/09/2022].

Responses:

 Thanks for the information. We added the specification in the description of participants. After confirmation from the supervisors of the teams, we ensured all the MATMs had no recorded mental health problems and several sleep problems before dispatchment. Regarding the base line of mental condition, we also confined MATMs’ subjective valuations of mood and sleep problems presented under current circumstances by prefixing the items with “in the past month” or “since being dispatched to the hospital”. Inevitably, some potential biases could happen when the study sampled at each time, however repeatedly or variedly. Therefore, contrasts between two-wave samples were made on the group levels and some inferences were made from regression analyses using the samples from the same wave. The separate regression analysis of our study would also reveal the interrelations between individual mental-health variations and sleep problems. Even if the individual had sleep issues prior to being dispatched, we anticipated that he or she would demonstrate a worse evaluation of the issues and higher scores on the relative items than the one without sleep problems.

 By the way, we thank you for providing the references but we only have a permitted access to the abstract. It would be a great favor to us if you could provide us the full text.     

Reviewer 3 Report

It is important to elaborate on the way risk is defined in the light of the 6 dimensions referred to in the paper so as to connect them to the results.

Author Response

Response:

  We added the concise concepts in the introduction for readers having a fast catch and the references for interests, and also added some COVID-19 studies for illustrating the relationship between risk perceptions and sleep problems.

Reviewer 4 Report

Introduction: please provide why did the authors select 2 hospitals and how different are there. why did not include only 1 hospital with a cohort studies for wave 1 and wave 2. 

Methods:  please provide the flow chart or figure that provide the readers to understand the protocol or methodology of the study. 

Discussion: please explain in the table 5. Please provide the reasons regarding wave 1 did not show significant in  negative emotion and physical risk but significances were shown in wave 2.

Author Response

  • Introduction: please provide why did the authors select 2 hospitalsand how different are there. why did not include only 1 hospital with a cohort studies for wave 1 and wave 2.

Response:

As we mentioned in the article- Our study compared the survey findings provided by the two MATM groups, who were the first to be sent to the lockdown city and its Square Cabin Hospitals, with a view to examining the changes in the connection. Especially, both groups encountered an un-expected and non-experienced challenges/stresses centered on the context of two similar significant and protracted lockdowns. Therefore, it would be not appliable to include only 1 hospital with a cohort studies for wave 1 and wave 2.

  • Methods:  please provide the flow chart or figure that provide the readers to understand the protocol or methodology of the study.

Response:

 Thanks for the constructive suggestion. We made a figure in the article.

  • Discussion: please explain in the table 5. Please provide the reasons regarding wave 1 did not show significant in negative emotion and physical risk but significances were shown in wave 2.

Response:

 We attributed the result to the effect of wave changes regardless of the group differences in age, gender, and occupation. We further explained in the discussion: our cohort of MATMs as the first-line warriors for public health were voluntary to the hardest-hit areas regardless of the threat of infection. S2 working in Shanghai cabin hospital showed more saliant pleasant feelings as a result of the spirits and encouragements. Negative emotions, on the other hand, were not dissipated. In comparison, S1 and S2 rated their terror differently (being afraid). Given the positive relationship between negative emotion and risk perceptions, we hypothesized that S1 experienced persistent fear of infection due to the virus's contagious nature, unknown transmission modes, close contact with patients, and infection occurring in their colleagues, whereas S2 had some knowledge of COVID-19 and adequate protections, appearing more psychologically prepared for the virus, but they were still unaware of the contagious and dangerous variants threaten by the physical safety. As a result, MATMs' perceptions of dangers and job preparations may benefit from maintaining a happy mood and controlling emotional responses.

Round 2

Reviewer 4 Report

Manuscript accepted.